# Enablers and barriers to community pharmacists' readiness to implement deprescribing of inappropriate medications for older adults in Qatar

**Marwa Elshazly**[⊙], **Sondus Jawad**[iD][⊙], **Ayesha Ahmed**[iD][⊙], **Hager ElGeed, Kazeem Babatunde Yusuff**[iD]*

Department of Clinical Pharmacy and Practice, College of Pharmacy, QU Health, Qatar University, Doha, Qatar

⊙ These authors contributed equally to this work.
* kyusuff@qu.edu.qa, yusuffkby@yahoo.co.uk

**Data Availability Statement:** All relevant data are within the manuscript.

**Funding:** The study was funded with an Undergraduate Research Experience Program

## Abstract

There is paucity of studies focused on the enablers and barriers to community pharmacists' readiness to deprescribe inappropriate medications for older adults in developing settings. The current study assessed the enablers and barriers to community pharmacists' readiness to implement deprescribing of inappropriate medications for older adults. A cross-sectional survey of 252 community pharmacists was conducted in Qatar with a pre-tested 24-item questionnaire developed with the theory of domain framework. Information about perceived enablers and barriers were elicited with a 5-point Likert-type scale. The response rate was 79.4% (200/252). The majority of the community pharmacists were females (54.5%), within the age range of 20–40 years (88.0%), had BSc / BPharm as the highest educational qualification (70.5%), were full-time employee (97.0%). The top-ranked enablers of community pharmacists' readiness to implement deprescribing were exposure to CPD on the use of deprescribing toolkits and algorithm (66%), interprofessional collaboration with physicians (60.5%) and shared electronic patient record (59.5%), and improved remuneration / reimbursement 58%). The top-ranked barriers were lack of access to patient records (70.5%), ineffective collaboration with physicians (66.5%), lack of time due to heavy workload (65%), regulatory framework that limit expansion of clinical roles (51%) and intense focus on sales target (49%). The top-ranked enablers of community pharmacists' readiness to implement deprescribing were exposure to CPD on the use of deprescribing toolkits and algorithm, interprofessional collaboration with physicians and shared electronic patient record. These findings bode well for the implementation of community pharmacists-led deprescribing of inappropriate medications for older adults in Qatar. However, a number of critical barriers were identified, and these will require institutional, regulatory and organizational interventions to improve readiness.

(UREP) award [UREP29-092-3-029] from the Qatar National Research Fund (a member of The Qatar Foundation). Open access funding was provided by the Qatar National Library. The contents of the manuscript are solely the responsibility of the authors, and the funders had no role in study design, data collection and analysis, decision to publish, or preparation of the manuscript.

**Competing interests:** The authors have declared that no competing interests exist.

## Introduction

Deprescribing is an important clinical tool that gained ascendancy in the last decade and has become crucial to reducing potentially harmful medication-related adverse events, and promoting appropriate and rational prescribing and use of medicines [1,2]. Indeed, several studies have shown that deprescribing has proven to be particularly useful for older adults who are often diagnosed with multiple chronic medical conditions that are usually managed with chronic multiple drug therapy, and which puts them at higher risk of adverse and unnecessary clinical and financial burdens [3–5]. The most critical determinant of the risk of exposure to adverse outcomes in older adults is the number of medicines prescribed [6], and studies have shown that the elderly population are particularly at higher risk due to ageing-related altered pharmacokinetics and pharmacodynamics factors [6–10]. In addition, older adults are at risk of harms from medications that are considered inappropriate and potentially harmful due to an inherent risk of exposure to adverse effects [6]. Therefore, deprescribing is a useful clinical tool that provides an opportunity for a conscientious review of the relevance and utility of the medications prescribed to older adults with a view to identify the medications that are no longer required or harmful, and should be discontinued or replaced with safer and more effective alternatives [11].

Published studies done in developed settings continue to showcase the positive outcomes of community pharmacists' participation in the deprescribing of inappropriate and potentially harmful medications for older adults. For instance, a systematic review by Buzancic et al, showed that community pharmacists' effective collaboration with physicians and patients in the implementation of deprescribing results in optimal therapeutic outcomes [12]. Furthermore, Martin et al, in a cluster randomized controlled trial among older adults in Canada reported that community pharmacists-led deprescribing resulted in greater discontinuation of inappropriate medications compared to usual care at 6 months (risk difference: 31% (95%CI, 23%-38%) [13]. In addition, Tannebaum et al., reported that community pharmacists-led deprescribing intervention resulted in patients' initiation of conversation with a physician or pharmacist about discontinuation or actual discontinuation of the use of benzodiazepine in older adults [14].

Despite the successful implementation of community pharmacists-led deprescribing intervention among older adults, a number of factors have been identified as enabling or militating against this intervention. The enabling factors include the use of tailored deprescribing guideline within a structured multidisciplinary framework, use of a multidisciplinary deprescribing model involving physicians, pharmacists and nurses, effective interprofessional collaboration and support, community pharmacists' role expansion into implementing deprescribing at the primary and long term care settings, effective engagement and involvement of patients and / or their relatives, adequate education on deprescribing and availability of financial incentive or re-imbursement [15–17]. The identified barriers to the implementation of community pharmacists-led deprescribing include lack of expertise and deprescribing awareness, poor self-efficacy and fear of consequences of deprescribing, poor communication and collaboration with patients and healthcare professionals, fragmentation of the healthcare system and poor information exchange, lack of incentives, insufficient resources including time and finance, and perception of an "established hierarchy" where physicians are regarded as senior to pharmacists and nurses [15–20].

Studies focused on community pharmacists-led deprescribing intervention and the factors enabling and / or militating against it were conducted mainly in developed settings. Literature search revealed the paucity of such studies in developing settings, including Qatar. Indeed, only one such study focused on the assessment of community pharmacists' knowledge of

deprescribing and their self-perceived enablers and barriers to providing the service in the United Arab Emirates was identified [21]. The paucity of studies focused on community pharmacists-led deprescribing intervention in developing settings is unsurprising as the provision of a deprescribing service guided by a structured framework is currently not within the scope of practice for community pharmacists generally in developing settings including in the Middle Eastern Gulf countries. Hence, a baseline assessment of the enablers and barriers to community pharmacists' implementation of deprescribing especially for a vulnerable group such as older adults seems like the appropriate starting point.

The current study was foregrounded by the Theoretical Domains Framework (TDF) V2 [21]. This is a theoretical framework that assesses the factors that may enable or militate against the successful implementation of an intervention focused on a specific desired practice change. The TDF consists of 14 domains developed from 128 theoretical constructs that were synthesized from varieties of behavioural change theories associated with implementation science and practice change [22–25]. The 14 domains of the TDF include knowledge; skills; social / professional identity; beliefs about capabilities; optimism; beliefs about consequences; reinforcement; intentions; goals; Memory, attention and decision processes; environmental context and resources; and social influences.

Hence, the current study may provide new significant perspectives that will not only add to global knowledge in the research area but may also provide insights that can be used to develop an appropriate institutional framework to guide the implementation of community pharmacists-led deprescribing for older adults in Qatar and other similar developing settings. The objective of the study was to identify the enablers and barriers to community pharmacists' readiness to implement the deprescribing inappropriate medication for older adults in Qatar.

## Methods

### Study design, setting and sampling

A cross-sectional survey of the enablers and barriers to community pharmacists' readiness to implement deprescribing for older adults was conducted between 01 November, 2023 and 20 January 2024 in Qatar, a Middle Eastern Gulf country with an estimated population of 2.73 million [26].

The study participants include a purposive sample of community pharmacists who were drawn from a sampling frame of all licensed community pharmacists in practice for at least a year in Qatar, and who were fluent in oral and written English. The Raosoft online calculator was used to obtain a minimum sample of 252 community pharmacists and an extra 10% was added to adjust for non-response. The parameters used for the sample size calculation include: target population of community pharmacists (562), alpha level (5%), confidence level (95%), and response distribution (50%).

### Questionnaire development and structure

The 14 domains of the TDF were grouped into 5 categories of deprescribing framework including 'knowledge', 'professional role and identity', 'beliefs about capabilities', 'environmental context and resources' and 'social influence'. This grouping is inherent in the TDF and was also based on an Irish study conducted by Henrich et al, that assessed the perceptions of healthcare professional about the barriers and enablers to deprescribing in long-term care setting [16]. The 5 categories of deprescribing framework were used to develop the initial draft of 30 items that was identified by the research team after a thorough review of relevant literature in the research area [15–21]. This was, followed by an iterative process involving an in-depth discussion of the appropriateness, relevance and validity of the items, and their mapping to the

deprescribing framework. This resulted in the final 23-item questionnaire comprising three sections including A (demographics = 10 items), B (Enablers = 7 items), and C (Barriers = 6 items). The questionnaire items were mapped to the deprescribing framework as follow: knowledge = 2, professional role and identity = 3, beliefs about capabilities = 2, environmental context and resources = 4 and social influence = 3. The content validity of the questionnaire was assessed by a panel of three experienced researchers in the research area. In addition, the internal consistency of the final questionnaire was determined with Cronbach alpha, and these were 0.82 and 0.75 for the sections B and C respectively. Lastly, the questionnaire was pre-tested to assess its clarity, completeness and relevance on a sample 10 community pharmacists, and minor modifications were made as necessary. The pre-test result was not included in the final results. The study participants were asked to rank their responses to the items in Section B and C on a 5-point Likert-type scale (highest = 5, high = 4, moderate = 3, low = 2, Lowest = 1). However, this response scale was re-coded into three categories of low (lowest + low), moderate and high (highest + high) to ease data analysis.

## Ethics approval

Ethics approval was obtained from QUIRB (Qatar University Institutional Review Board) (QU-IRB 1906-E/23, August 30, 2023).

## Data collection and analysis

Three research assistants (students) collected the data with the validated and pre-tested 23-item questionnaire. The data collection procedure was standardized to minimize variation. The questionnaires were delivered to the respondents at their workplace, and detailed information about the purpose and anticipated benefits of the study were provided in a separate informed consent form. The respondents who agreed to participate in the study signed the written informed consent form before the start of data collection, and they were all informed of their right to decline participation at any point in the study. Completed questionnaires were collected as soon as possible but not exceeding 5 days after distribution. Reminders were sent through telephone calls or emails to those who did not return completed questionnaires within one week. The IBM SPSS (Statistics for Windows, Version 29.0. Armonk, NY: IBM Corp.) software was used for data analysis. Descriptive statistics such as median (IQR) was used for continuous data with non-normal distribution while frequency (%) was used for categorical data. The alpha level for significance was set at $p \leq 0.05$.

## Results

Of the 252 respondents who consented, two hundred completed the questionnaires (response rate, 79.4%). The demographic characteristics of the respondents showed that the majority were females (54.5%; 109/200), within the age range of 20–40 years (88.0%; 176/200), had BSc / BPharm as the highest educational qualification (70.5%; 141/200), were full-time employee (97.0%; 194/200), and consisted mainly of 5 nationalities (91.0%; 182/200) including Indian (40.5%), Egyptian (22.5%), Jordanian (22%) and Sudanese (6%). Furthermore, the majority of the community pharmacists had 1–5 years of practice experience (51.5%; 103/200) and previous hospital pharmacy experience (55%; 110/200) (Table 1). However, only 49% (98/200) reported exposure to deprescribing during undergraduate training. In addition, the majority of respondents reported not attending any CPD related to deprescribing in the past 5 years (54%; 108/200). However, the median (IQR) score for the willingness to complete a CPD program on deprescribing was 4 (2) (maximum = 5) (Table 1).

**Table 1. Community pharmacists' demographic and practice characteristics (N = 200).**

| Item | n (%) |
|---|---|
| **Gender** | |
| Male | 91 (45.5) |
| Female | 109 (54.5) |
| **Age group (years)** | |
| 20–30 | 102 (51.0) |
| 31–40 | 74 (37.0) |
| 41–50 | 24 (12.0) |
| **Nationality** | |
| Indian | 81 (40.5) |
| Egyptian | 45 (22.5) |
| Jordanian | 44 (22.0) |
| Sudanese | 12 (6.0) |
| Pakistani | 7 (3.5) |
| Filipino | 4 (2.0) |
| Syrian | 3 (1.5) |
| Saudi | 2 (1.0) |
| Palestinian | 1 (0.5) |
| Ghana | 1 (0.5) |
| **Highest educational Level** | |
| BSc/BPharm | 141 (70.5) |
| MSc / MPharm | 33 (16.5) |
| PharmD | 26 (13.0) |
| **Employment status** | |
| Full time | 194 (97.0) |
| Part time | 6 (3.0) |
| **Years of experience as a community pharmacist** | |
| 1–5 years | 103 (51.5) |
| 6–10 years | 57 (28.5) |
| >10 years | 40 (20.0) |
| **Previous hospital pharmacy experience** | |
| Yes | 110 (55.0) |
| No | 90 (45.0) |
| **Exposure to deprescribing in undergraduate training** | |
| Yes | 98 (49.0) |
| No | 102 (51.0) |
| **Attended CPD related to deprescribing in past 5 years** | |
| None | 108 (54) |
| 1–2 | 56 (28.0) |
| 3–4 | 18 (9.0) |
| >4 | 18 9) |
| **Willingness to complete CPD on deprescribing (Median (IQR))** | 4 (3, 5) |

The enablers and barriers to community pharmacists' readiness to implement deprescribing for older adults in Qatar is shown in Table 2. The top-ranked enablers identified by the majority of respondents include continuous CPD on the use of deprescribing toolkits and algorithm (66%; 132/200), established forum for an effective interprofessional collaboration with physicians (60.5%; 121/200), availability of shared electronic patient record (59.5%; 119/

**Table 2. Enablers and barriers to the community pharmacists' readiness to implement deprescribing for older adults in Qatar (N = 200).**

| No | Questionnaire items | Low n (%) | Moderate n (%) | High n (%) | Median (IQR) |
|---|---|---|---|---|---|
| | **Enablers** | | | | |
| 1 | CPD on deprescribing toolkits and algorithm and how to use them. | 15 (7.5) | 53 (26.5) | 132 (66.0) | 4 (3, 5) * |
| 2 | Established forum for effective interprofessional collaboration with physicians. | 27 (13.5) | 52 (26.0) | 121 (60.5) | 4 (3, 5) * |
| 3 | Availability of a shared electronic patient record for physicians and community pharmacists. | 34 (17.0) | 47 (23.5) | 119 (59.5) | 4 (3, 5) * |
| 4 | Regulatory permission to community pharmacists in Qatar to provide clinically-oriented services such as deprescribing | 41 (20.5) | 62 (31.0) | 97 (48.5) | 3 (3, 4) |
| 5 | Improved remuneration for community pharmacists to provide clinically-oriented services including deprescribing. | 30 (15.0) | 54 (27.0) | 116 (58.0) | 4 (3, 5) * |
| 6 | Public education for patients to know the benefits of deprescribing. | 44 (22.0) | 55 (27.5) | 101 (50.5) | 4 (3, 5) * |
| 7 | Opportunities exist to deprescribe inappropriate medications in my practice environment. | 21 (10.5) | 68 (34.0) | 111 (55.5) | 4 (3, 5) * |
| | **Barriers** | | | | |
| 1 | Lack of access to a detailed patient medical / medication history. | 21 910.5) | 38 (19.0) | 141 (70.5) | 4 (3, 5) * |
| 2 | Workload pressure leading to lack of time to review prescription for deprescribing opportunities. | 32 (16.0) | 38 (19.0) | 130 (65.0) | 4 (3, 5) * |
| 3 | Limited knowledge of the deprescribing toolkits and algorithm and how to use them. | 35 (17.5) | 70 (35.0) | 95 (47.5) | 3 (3, 4) |
| 4 | Lack of effective collaborative working relationship with physicians. | 32 (16.0) | 35 (17.5) | 133 (66.5) | 4 (3, 5) * |
| 5 | Intense focus on meeting given organizational sales / financial target | 34 (17.0) | 68 (34.0) | 98 (49.0) | 3 (3, 4) |
| 6 | Detailed medication usage review is not currently a key part of community pharmacists' job description in Qatar. | 57 (28.5) | 49 (24.5) | 94 (47.0) | 3 (3, 4) |
| 7 | Current pharmacy regulatory framework limits the range of clinically-related services that could be provided by community pharmacists in Qatar. | 40 (20.0) | 58 (29.0) | 102 (51.0) | 4 (3, 5) * |

200), improved remuneration for community pharmacists (58%; 116/200), opportunities to deprescribe inappropriate medications in practice (55.5%; 111/200), and public education on the benefits of deprescribing inappropriate medications (50.5%; 101/200) (Table 2).

The top-ranked barriers by the majority of community pharmacists were lack of access to patient records (70.5%; 141/200), lack of an effective interprofessional collaboration (66.5%; 133/200), lack of time due to heavy workload (65%; 130/200) and current regulatory framework for community pharmacy practice in Qatar (51%; 102/200) and intense focus on meeting given sales target (49%; 98/200).

## Discussion

The identification of opportunities to attend CPD programs focused on the effective use of deprescribing toolkits and algorithm as one of the top-ranked enablers seems unsurprising as only about half of the respondents reported exposure to deprescribing in their undergraduate training, and attendance of CPD program related to deprescribing in the past five years. This finding is consistent with that of Cheong et al, and Heinrich et al, who reported education about deprescribing resources and how to use them in practice as a top-ranked facilitator of initiative to enhance community pharmacists' capacity to provide deprescribing service [27,28]. Therefore, the identification of attendance of CPD programs focused on the effective

used of deprescribing toolkits and algorithm bodes well for enhancing community pharmacists' readiness for this role expansion within the healthcare system. This is a plausible proposition as the majority of respondents in the current study reported a high willingness to complete a CPD program focused on deprescribing of inappropriate medications for older adults in Qatar.

Furthermore, the identification of the availability of a forum for an effective interprofessional collaboration with physicians and a shared patient electronic record and improved remunerations for community pharmacists as top-ranked enablers align with trends reported by published studies from developed settings [15,16,27]. These are propositions that will enhance community pharmacists' readiness to effectively implement a clinical service such a deprescribing in Qatar. Several studies have documented the benefits associated with the use of an effective interprofessional collaborative model in enhancing effective provision of deprescribing service by community pharmacists [29,30]. In addition, the availability of a shared electronic patient record will improve community pharmacists' access to critical information needed to identify and deprescribe inappropriate medications for older adults [31,32]. The identification of improved remuneration package for community pharmacists as one of the top-ranked is similar to the findings reported by Heinrich et al, [27] and Jokanovic et al. [33] This is unsurprising as the positive impact of an improved remuneration package on productivity, job satisfaction and readiness to complete assigned task are well documented [34,35].

The identification of existing opportunities to deprescribe inappropriate medications in practice by a majority of respondents is probably a good pointer of the current burden of the inappropriate medications that older adults are exposed to in Qatar. In addition, community pharmacists' identification of public education of the benefit of providing deprescribing service especially at the community level as an enabler is probably an indicator of their commitment to provide this service. Perhaps, such a public enlightenment program may enhance the demand for such a service at the community level where community pharmacies abound [36].

Furthermore, it was unsurprising that lack of access to patient record was one of the top-ranked barriers identify by the respondents. The critical roles that access to detailed medical and medication history play in enhancing the ability to analyze, identify and resolve clinical problems are well documented [32]. In addition, the identification of lack of existing effective interprofessional collaboration and lack of time due to heavy workload as key barriers to the implementation of community pharmacists-led deprescribing are well documented [21,27,31]. Therefore, a holistic intervention targeted towards mitigating these barriers are warranted. Furthermore, the identification of the regulatory framework currently guiding community pharmacy practice in Qatar as a key barrier to the implementation of deprescribing in practice is being reported for the first time. This is because the range of clinically-oriented services that community pharmacists are legally permitted to provide are limited and not in tandem with the expanding needs at the community level. Therefore, there is a clear need for a review of the community pharmacy regulatory framework to ensure that it is in tandem with the expanding roles of community pharmacists, and meet the changing societal needs in the contemporary times. Indeed, the review of the regulatory framework is the crucial first step that must be undertaken in developing a scheme to expand the range of clinically-oriented services, including deprescribing, that community pharmacists will be able to provide to patients in Qatar.

Lastly, the identification of community pharmacists' focus on meeting organizational sales target as a top-ranked barrier to readiness to implement deprescribing is being reported for the first time, and seems to suggest a dominant focus on the business side rather clinical practice in the community pharmacy sector in Qatar. This is unsurprising as organizational policies and goals, which are often shaped by organizational vision, are crucial determinants of job-related behavior, and seem to be an important factor to consider in developing an

institutional framework for the implementation of community pharmacists-led deprescribing. This seems plausible as previous studies have reported that re-imbursement or remuneration of community pharmacists based on the number of prescriptions filled or medications dispensed or sold could be become a financial disincentive for implementing deprescribing in practice [16,20,37].

## Strengths and limitations

To best of our knowledge, the current study is the first from a developing setting to foreground the assessment of enablers and barriers to the readiness of community pharmacists to implement deprescribing with the TDF, a widely used theoretical framework in implementation research. However, the study findings may be limited by a few factors including the use of a non-probability sampling as the participants were drawn mainly from the chain pharmacies. However, chain pharmacies constitute about 75% of the community pharmacies in Qatar [38]. Hence, the reported findings probably approximate the reality on the ground. Furthermore, the use quantitative survey design may have increased the risk of social desirability bias as the participants who responded may have done so in a favorable manner. However, this appeared not be the case as a sizeable proportion of top-ranked barriers were identified by the respondents.

## Conclusions

The top-ranked enablers of community pharmacists' readiness to implement deprescribing in practice were exposure to CPD on the use of deprescribing toolkits and algorithm, interprofessional collaboration with physicians and shared electronic patient record, and improved remuneration / re-imbursement. These findings bode well for the implementation of community pharmacists-led deprescribing of inappropriate medications for older adults in Qatar. However, a number of critical barriers were identified, and these will require institutional, regulatory and organizational interventions to improve readiness. The top-ranked barriers were lack of access to patient records, ineffective collaboration with physicians, lack of time due to heavy workload, regulatory framework that limit expansion of clinical roles, and intense focus on sales target.

## Author Contributions

**Conceptualization:** Marwa Elshazly, Sondus Jawad, Ayesha Ahmed, Hager ElGeed, Kazeem Babatunde Yusuff.

**Data curation:** Marwa Elshazly, Sondus Jawad, Ayesha Ahmed, Kazeem Babatunde Yusuff.

**Formal analysis:** Marwa Elshazly, Sondus Jawad, Ayesha Ahmed, Hager ElGeed, Kazeem Babatunde Yusuff.

**Funding acquisition:** Kazeem Babatunde Yusuff.

**Methodology:** Marwa Elshazly, Sondus Jawad, Ayesha Ahmed, Kazeem Babatunde Yusuff.

**Project administration:** Kazeem Babatunde Yusuff.

**Software:** Kazeem Babatunde Yusuff.

**Supervision:** Kazeem Babatunde Yusuff.

**Validation:** Kazeem Babatunde Yusuff.

**Visualization:** Hager ElGeed, Kazeem Babatunde Yusuff.

**Writing – original draft:** Marwa Elshazly, Sondus Jawad, Ayesha Ahmed, Hager ElGeed, Kazeem Babatunde Yusuff.

**Writing – review & editing:** Marwa Elshazly, Sondus Jawad, Ayesha Ahmed, Hager ElGeed, Kazeem Babatunde Yusuff.

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
