## [Decision Letter · Decision Letter 0]

20 Sep 2024

PONE-D-24-26224Enablers and barriers to community pharmacists’ readiness to implement deprescribing of inappropriate medications for older adults in a developing settingPLOS ONE

Dear Dr. Yusuff,

Thank you for submitting your manuscript to PLOS ONE. After careful consideration, we feel that it has merit but does not fully meet PLOS ONE’s publication criteria as it currently stands. Therefore, we invite you to submit a revised version of the manuscript that addresses the points raised during the review process.

**Both reviewers have provided comments to improve the clarity of your work particularly regarding the decision making rationale for the study and data presentation. **

We look forward to receiving your revised manuscript.

Kind regards,

Jenny Wilkinson, PhD

Academic Editor

PLOS ONE

**Journal Requirements:**

KBY

Undergraduate Research Experience Program (UREP) award [UREP29-092-3-029] from the Qatar National Research Fund (a member of The Qatar Foundation). The contents are solely the responsibility of the authors.

Reviewers' comments:

Reviewer's Responses to Questions

**Comments to the Author**

1. Is the manuscript technically sound, and do the data support the conclusions?

Reviewer #1: Partly

Reviewer #2: Yes

2. Has the statistical analysis been performed appropriately and rigorously? 

Reviewer #1: I Don't Know

Reviewer #2: Yes

3. Have the authors made all data underlying the findings in their manuscript fully available?

Reviewer #1: No

Reviewer #2: No

4. Is the manuscript presented in an intelligible fashion and written in standard English?

Reviewer #1: Yes

Reviewer #2: Yes

5. Review Comments to the Author

**Reviewer #1: **The manuscript as presented could benefit from additional information, rationale to support decision-making and clarity. These follow in summary and more detail in peer review report.

1. Amend title to include Qatar

2. Offer definition of deprescribing for this study Lines 57-59. Deprescribing is identified as “… an important clinical tool…” but for clarity, it would be helpful to offer a definition or detailed description of deprescribing as used in this study.

3.Lines 85-88 would benefit from each key assertion being individually referenced

4. Lines 102-104 It can be helpful to identify the search criteria utilised

5. Lines 112-117 It could assist readers to identify additional information such as the fourteen domains and those especially relevant to this study, for example

6.Lines 140-144 ? Please explain the research team’s decision-making on this point. Please describe the fourteen domains and the theoretical background as it informs this study.

7. Lines 170-173 This section could be enhanced by some details about the research assistants since they seem to be the “face” of this study to pharmacist participants.

8. Analysis section - In a manuscript, it is beneficial to identify those functions within SPSS that were used to analyse the data, similarly any recoding of values such as for negatively phrased items. More detailed information is needed to understand the results obtained and the conclusions drawn.

9.study participants were asked to rank their responses to the items in Section B and C on a 5-point Likert-type scale (highest =5, high = 4, moderate = 3, low = 2, Lowest = 1). However, in Results section Table 2, the columns are labelled,

Low

n (%) Moderate

n (%) High

n (%) Median (IQR)

If additional analyses or re-coding were undertaken to result in these three categories, that needs to be clearly detailed.

10. Apparent repetition Lines 173-176 AND Lines 165-167 - Both state “The respondents who agreed to participate in the study signed the written informed consent form before the start of data collection, and they were all informed of their right to decline participation at any point in the study.”

11. Results Additional detail would be helpful to understand the results of the study and its outcomes.

12. Additional guidance is available in this article, “A Consensus-Based Checklist for Reporting of Survey Studies (CROSS)” in which the reporting guidelines are found in the supplementary material – supplement 3

13.The discussion benefits from being better contextualised by discussing recent publications centred on Qatar

14. Limitations - it was not evident in the manuscript which locations were sampling towns or regions as differences in perceptions, experiences and resources might be anticipated in various locations

15. The the link between methodology (survey) and favourable responses is not clearly explained but benefits from being so.

16.This reference (number 36) appears to be inconsistent with Vancouver referencing:

Gerlach N, Michiels-Corsten M, Viniol A, et al. Professional roles of general practitioners, community pharmacists and specialist providers in collaborative medication deprescribing—a qualitative study. BMC Fam Pract. 2020/09/06 ed. 2020 Sep 4; 21(1):183.

17. Please review all references for consistency with Vancouver format.

**Reviewer #2: **The manuscript is good but I think the following points should be addressed:

1- rewrite the title as the title is not clear such using facilitators instead of enablers

2- I prefer to attach the developed questionnaire in the appendix or supplement file

3- I think also mention how did you summarize answers from five choices to three choice (table 2) in the methodology section after line 161-162

4- I think mention the current practice in Qatar, it is manly traditional practice or there is/are some advance practice such as reviewing medication, providing vaccination, prescribing authorizes, etc... will be beneficial and strengthen the study

6. PLOS authors have the option to publish the peer review history of their article (what does this mean?). If published, this will include your full peer review and any attached files.

Reviewer #1: No

Reviewer #2: **Yes: **Mohammed Abdullah Kubas

---

## [Author Response · Author response to Decision Letter 0]

29 Oct 2024

29 October 2024

The Editor-In-Chief

PLOS ONE

Dear Sir,

Re: Manuscript #PONE-D-24-26224 – “Enablers and barriers to community pharmacists’ readiness to implement deprescribing of inappropriate medications for older adults in a developing setting”

Our sincere thanks for the opportunity to revise the manuscript #PONE-D-24-26224, titled “Enablers and barriers to community pharmacists’ readiness to implement deprescribing of inappropriate medications for older adults in a developing setting” which is under your consideration for publication in the PLOS ONE. We thank the reviewers for the insightful comments and useful suggestions and we have revised the manuscript accordingly. Please find stated below our point-by-point response to the reviewers’ comments. We have also revised the manuscript in accordance with the editor’s comments.

EDITOR’S COMMENTS

Journal Requirements:

1. When submitting your revision, we need you to address these additional requirements. Please ensure that your manuscript meets PLOS ONE's style requirements, including those for file naming. The PLOS ONE style templates can be found at https://journals.plos.org/plosone/s/file?id=wjVg/PLOSOne_formatting_sample_main_body.pdf and

Response: The corrections have been done in accordance with specifications stated in the PLOS ONE style template

KBY

Undergraduate Research Experience Program (UREP) award [UREP29-092-3-029] from the Qatar National Research Fund (a member of The Qatar Foundation). The contents are solely the responsibility of the authors. Please state what role the funders took in the study. If the funders had no role, please state: "The funders had no role in study design, data collection and analysis, decision to publish, or preparation of the manuscript." If this statement is not correct you must amend it as needed. Please include this amended Role of Funder statement in your cover letter; we will change the online submission form on your behalf.

Response: The financial disclosure statement has been amended as recommended, and also included in the revised cover letter.

Response: The abstract has been amended as recommended.

REVIEWER’S COMMENTS

Reviewer #1: The manuscript as presented could benefit from additional information, rationale to support decision-making and clarity. These follow in summary and more detail in peer review report.

Response: We are truly grateful for the valuable comments offered by the reviewer and the time spent to provide these useful feedback. We value the suggested corrections proposed by the reviewer and we are convinced it can only improve the scholarly value of the manuscript.

• Comment-1: Amend title to include Qatar.

Response-1: We thank the reviewer for this suggestion and we totally concur with the premise of the suggested correction. The title has been revised to include “Qatar” [Pg 1, line 2].

• Comment-2: Currently those in Qatar older than sixty years of age constitute 3.6% of the population though that is predicted to increase to approximately 20.3% by 2050, which is in 26 years’ time, so lacking apparent immediate urgency. So, seek to identify the significance or implications of undertaking this study now in 2024. There is time available to review and amend pre-registration qualifications offered in Qatar by adding or increasing medicines simplification, quality use of medicines and de-prescribing content. In addition, there is time to develop continuing professional development materials, and propose credentialing or recognized qualifications to registering authorities.

• Response-2: We thank the reviewer for the important observation and we see the points raised. However, we are of the opinion that the potential clinical and financial burden associated with the harms inherent in the use of inappropriate medications in elderly population is better avoided by provision of a clinical service such as deprescribing especially at the primary care level where community pharmacists are easily accessible. In addition, 3.6% of the current population translates to about 120,000 elderly patients in Qatar (Estimated pop is about 3million). Hence, there is a lot of potential inherent benefits associated with the deployment of a care-enhancing clinical service such as deprescribing of inappropriate medications in one of the most vulnerable patient group such as the elderly. In addition, it is generally well established that early deployment of interventions focused improving patient outcomes and strengthening the health system often result lasting positive impact on public health. 

• Comment-3: KEY WORDS: Currently- Community pharmacists, deprescribing, older adults, enablers, barriers. There is mention of training needs, education enhancement in the manuscript so perhaps adding : curriculum review, continuing professional development.

Response-3: This is an excellent suggestion and we sincerely thank the reviewer. The revision has been done as recommended [Pg 3, line 47-48].

• Comment-4: Line 31 The current study assessed determined… It is possible to use one or other word.

• Response-4: Heartfelt thanks to the reviewer. This was an unintended error and it has been corrected [Pg 2, line 24].

• Comment-5: Lines 35-36 “Enablers and barriers were assessed with a 5-point Likert-type scale” . Something such as - Information about perceptions of enablers and barriers were elicited using a 5-point Likert scale - This provides additional clarity as to the purpose and what was disclosed by respondents.

• Response-5: Many thanks to the reviewer for the suggestion. The corrections have been done [Pg 2, line 28].

• Comment-6: Lines 57-59. Deprescribing is identified as “… an important clinical tool…” but for clarity, it would be helpful to offer a definition or detailed description of deprescribing as used in this study. Although deprescribing as a term was stated to be first used in the English language health literature in 2003 in an Australian hospital pharmacy journal, over subsequent years the term has been customised, for example in patients taking cardiovascular medicines only, and the process to some extent modified(Reeve, E., Gnjidic, D., Long, J., & Hilmer, S. (2015). A systematic review of the emerging definition of 'deprescribing' with network analysis: implications for future research and clinical practice. British journal of clinical pharmacology, 80(6), 1254–1268. https://doi.org/10.1111/bcp.12732) , so clarity in this study’s use is beneficial

• Response-6: Heartfelt thanks to the reviewer for this valuable comment. We agree that a definition of deprescribing will further enhance clarity. In fact, we respectfully submit that we have already done this in the last sentence of the first paragraph of the Introduction section. However, this sentence has been revised to enhance its clarity as recommended by the reviewer. In addition, we have also added one of the references kindly suggested by the reviewer because we think its definition of deprescribing seems to have been captured in line 61-65 [Pg 3-4] 

• Comment-7: Lines 63-65 assert that the greatest risk to the elderly from medication lies in the number of medicines, and certainly, the more medicines taken, the more there can be a risk of drug-drug, drug-herb, drug-disease state interactions, but there are also some medicines that have an inherent risk of adverse effects in the elderly, even with a lesser number of medications taken. These include NSAIDS, medicines with an anticholinergic effect, some diuretics, antihypertensives and others (https://www.msdmanuals.com/en au/professional/geriatrics/drug-therapy-in-older-adults/drug-categories-of-concern-in-older-adults).

• Response-7: Sincere thanks to the reviewer for this excellent comment. We concur with the reviewer point and we think this assertion has already been captured in the manuscript [Line 61-65]. We were conscious of balancing comprehensiveness with conciseness to minimize redundancies.

• Comment-8 Lines 76-78- Currently, “Furthermore, Martin et al, in a cluster randomized controlled trial among older adults in Canada reported that community pharmacists-led deprescribing resulted in greater discontinuation of inappropriate medications.” So, what is the comparator- Greater than prescribers? Greater than the patient’s requests? This benefits from additional clarity as to the insights relevant to or which informed this study.

Response-8: Sincere thanks for this important comment. The correction has been done to enhance clarity [Line 73-74].

• Comment-9: Lines 85-88. – Currently, “The enabling factors include the use of tailored deprescribing guideline within a structured multidisciplinary framework, use of a multidisciplinary deprescribing model involving physicians, pharmacists and nurses, effective interprofessional collaboration…”. These statements are referenced in the paragraph with references 14-16. Readers seeking more information perhaps of an unfamiliar term such as “a structured multidisciplinary framework” or an unfamiliar procedure such as “… use of a multidisciplinary deprescribing model…” would benefit from each key assertion being individually referenced, especially as these are identified as enabling factors.

Response-9: Many thanks to the reviewer and we see the point. However, for the sake of brevity, we cited the references at the end of the sentence. We think this is appropriate and consistent with the established norm in the scholarly world. 

Comment-10 - Lines 92-98. Likewise, the benefit of additional clarity for the barriers.

Response-10 - Many thanks to the reviewer and we see the point. However, for the sake of brevity, we cited the references at the end of the sentence. We think this is appropriate and consistent with the established norm in the scholarly world. 

• Comment-11: Lines 102-104. The manuscript states, “Indeed, only one such study focused on the assessment of community pharmacists’ knowledge of deprescribing and their self perceived enablers and barriers to providing the service in the United Arab Emirates was identified.” It can be helpful to identify the search criteria utilised as a quick search identified four studies that may have been helpful, two studies in 2024 in Nigeria (exploring barriers and enablers), one in Malaysia in 2023(deprescribing) and also in 2023 the study cited in reference 20 (key to promoting deprescribing). 

• Response-11: Many thanks for the interesting suggestion. The studies alluded to in the review comments that were conducted in Nigeria and Malaysia were done among HCPs including physicians, hospital pharmacists, nurses practicing in secondary and tertiary care settings. Our study was focused mainly on community pharmacists.

• Comment-12: Lines 112-117. The manuscript states, “The current study was foregrounded by the Theoretical Domains Framework (TDF) V2 [21]. This is a theoretical framework that assesses the factors that may enable or militate against the successful implementation of an intervention focused on a specific desired practice change. The TDF consists of fourteen domains developed from 128 theoretical constructs that were synthesized from varieties of behavioural change theories associated with implementation science and practice change…” It could assist readers to identify additional information such as the fourteen domains and those especially relevant to this study, for example.

• Response-12: Great comment by the reviewer and we are thankful for this. However. We respectfully submit that the TDF was adequately referenced. However, the mansucript has been revised to provide a summary of the 14 domains of the framework [line 112-116]. We thought this suffices for any interested readers who desire more detailed information about the framework. However

Atkins L, Francis J, Islam R, et al. A guide to using the theoretical domains framework of behaviour change to investigate implementation problems. Implement Sci. 2017;12:77.

Davis R, Campbell R, Hildon Z, Hobbs L, Michie S. Theories of behaviour and behaviour change across the social and behavioural sciences: a scoping review. HealPsychol Rev. 2015;9:323–344. 

Cadogan C, Ryan C, Francis J, et al. Improving appropriate polypharmacy for older people in primary care: selecting components of an evidence-based intervention to target prescribing and dispensing. Implement Sci. 2015;10:161. 

Alqubaisi M, Tonna A, Strath A, Stewart D. Quantifying behavioural determinants relating to health professional reporting of medication errors: a cross-sectional survey using the theoretical domains framework. Eur J Clin Pharmacol. 2016;72:1401–1411.

• Comment-13: Lines 140-144. Please see comment above seeking the fourteen domains and those most relevant to this study. It is stated the fourteen domains were grouped into 5 categories, were these grouped by the research team or is this grouping inherent in the TDF? It goes on to say this grouping was influenced by an Irish study but would an Irish study and a Qatari study benefit from the same/similar categorisation? Please explain the research team’s decision-making on this point. Please describe the fourteen domains and the theoretical background as it informs this study. In which geographical region was the framework developed? How does it apply to Qatar.

Response-13: Many thanks for this suggestion. The grouping into 5 domains is inherent in the TDF and it was also based on a published Irish study, and we have revised the manuscript to clarify this. The TDF framework is a validated and well documented theoretical framework that has been in several published studies focused on understanding factors associated with behavioral change and implementation science. Relevant references regarding the details of how the framework was developed and validated have been provided in the manuscript. The grouping into 5 domains is inherent in the TDF and it was also based on a published Irish study. The research team adopted these domains because of their well-documented impact on behavioral change irrespective of settings. Additional details regarding the research team decision-making have been provided in line 142-163. 

• Comment -14: Lines 159-160. For clarity please, does this mean that the responses from the pre-test sample were not included in the dataset of the study for analysis? What happened to them?

• Response-14: The pre-test results were discarded and this is in accordance with the established standard in Quantitative survey research.

• Comment-15: Lines 170-173. The three research assistants, were these individuals, students, administrative support people or pharmacists? This section could be enhanced by some details about the research assistants since they seem to be the “face” of this study to pharmacist participants. This is a concern because response rates for participation and completion can vary with the characteristics of the person providing the questionnaire. Back in 2002, it was established that even the colour of the paper a printed questionnaire was printed on made a difference, Etter and colleagues, for example, established that pink paper increased response rates by 12%, so provision of detail is important.

• Response-15: Many thanks for this important observation. The research assistants were students [3] and the data collection procedure was standardized to minimize any potential bias alluded to in the review comment [line 169-173].

• Comment-16: Line 173. The detailed “information about the purpose and anticipated benefits of the study,” how was the information provided? Verbally, written, link to study website? Additional information will assist readers as they consider their own studies.

• Response-16: Heartfelt thanks to the reviewer for this comment. The information about the purpose and anticipated benefits of the study was iincluded in the participant consent form given to the respondents to read and sign before the start of data collection. The manuscript has been revi

---

## [Decision Letter · Decision Letter 1]

21 Nov 2024

PONE-D-24-26224R1Enablers and barriers to community pharmacists’ readiness to implement deprescribing of inappropriate medications for older adults in QatarPLOS ONE

Dear Dr. Yusuff,

Thank you for submitting your manuscript to PLOS ONE. After careful consideration, we feel that it has merit but does not fully meet PLOS ONE’s publication criteria as it currently stands. Therefore, we invite you to submit a revised version of the manuscript that addresses the points raised during the review process.

 Thank you for your responses and revisions. The original reviewers have evaluated your responses and have asked for some further minor revisions to improve clarity of your work.

We look forward to receiving your revised manuscript.

Kind regards,

Jenny Wilkinson, PhD

Academic Editor

PLOS ONE

Journal Requirements:

Reviewers' comments:

Reviewer's Responses to Questions

**Comments to the Author**

1. If the authors have adequately addressed your comments raised in a previous round of review and you feel that this manuscript is now acceptable for publication, you may indicate that here to bypass the “Comments to the Author” section, enter your conflict of interest statement in the “Confidential to Editor” section, and submit your "Accept" recommendation.

Reviewer #1: (No Response)

Reviewer #2: All comments have been addressed

2. Is the manuscript technically sound, and do the data support the conclusions?

Reviewer #1: Partly

Reviewer #2: Yes

3. Has the statistical analysis been performed appropriately and rigorously? 

Reviewer #1: I Don't Know

Reviewer #2: Yes

4. Have the authors made all data underlying the findings in their manuscript fully available?

Reviewer #1: No

Reviewer #2: Yes

5. Is the manuscript presented in an intelligible fashion and written in standard English?

Reviewer #1: Yes

Reviewer #2: Yes

6. Review Comments to the Author

Reviewer #1: Thank you for submitting your revised manuscript. To my perception the changes made have enhanced the clarity and communication of the manuscript. There are also opportunities for additional clarity and enhanced communication of your research, the decision-making inherent to the research and the outcomes. In particular, it is important to meet the journal's expectations for key sections such as the introduction which provides sufficient context and background for readers unfamiliar with the study site to understand the setting, professional practice in that setting, and, issues that may benefit from exploration and dissemination. Then the next key component - the methodology which benefits from sufficient information and justification for decisions made such as the choice of function within a data analysis program to inform the reader why and for which purpose that analysis plan has been effected. Lastly, the discussion provides some context within which the results of the study can be situated and considered - so it may be what is specific about pharmacy and pharmacy practice in Qatar.

PLOS ONE Review 2

TITLE: Enablers and barriers to community pharmacists’ readiness to implement deprescribing of inappropriate medications for older adults in Qatar.

RELEVANCE OR SIGNIFICANCE: Currently those in Qatar older than sixty years of age constitute 3.6% of the population though that is predicted to increase to approximately 20.3% by 2050, which is in 26 year’s time, so lacking apparent immediate urgency. So,please seek to firmly and clearly identify the significance or implications of undertaking this study now in 2024 - Why is it needed? Why now? What has to change and what else has to happen for the activity on which this research is focussed to come into effect. In my experience, often authors may be well aware of what is happening in their setting, what could be desirable and will of the pharmacist population to provide a proposed service or modify a standard procedure, for example. However, readers from outside the setting are often quite unfamiliar with a study's setting, and in lieu of more information may wonder if it is much the same as in t he reader's setting. In my experience providing context, information, including some statistics will assist the reader the view the study within the actual study setting.

INTRODUCTION: PLOS ONE expectations for introductory material is for: provision of sufficient background and context of the manuscript such that readers are able to understand the purpose and significance of the study; clarity in identifying the issue addressed and why it is problematic or necessary; concluding with a brief description of the overall aim and whether/how that was/was not achieved. Comments on this section of this manuscript follow.

Some were requested previously but not clarified or reviewed to reflect the information/request:

Issue Lines 57-59. Deprescribing is identified as “… an important clinical tool…” but for clarity, it would be helpful to offer a definition or detailed description of deprescribing as used in this study.

Authors response:

Response-6: Heartfelt thanks to the reviewer for this valuable comment. We agree that a definition of deprescribing will further enhance clarity. In fact, we respectfully submit that we have already done this in the last sentence of the first paragraph of the Introduction section. However, this sentence has been revised to enhance its clarity as recommended by the reviewer. In addition, we have also added one of the references kindly suggested by the reviewer because we think its definition of deprescribing seems to have been captured in line 61-65 [Pg 3-4]

The lines referred to have content as follows:

provides an opportunity for a dispassionate review of the relevance and utility of the medications prescribed to older adults with a view to identify the medications that are no longer required or harmful, and should be discontinued or replaced with safer and more effective alternatives.

Issue - A little more clarity would beneficial - what is a dispassionate review? What benefit arises from a dispassionate view? Why not a standard protocol? Why not take into consideration the patient’s perceptions of their medications or issues they may be having? Further, the key is to identify medications that are no longer required - for which reasons? duplication of therapy? continuation of medication to manage an acute condition? removal of a medication that manages an adverse effect of another medication taken by the patient?

Issue- benefit of additional clarity as to why deprescribing needs to be added, legally, to community pharmacists scope, and why the timing since hospital pharmacists have and are trialling a deprescribing initiative- how is it that deprescribing is within scope for hospital pharmacists in Qatar but not community pharmacists?

Response-2: We thank the reviewer for the important observation and we see the points raised. However, we are of the opinion that the potential clinical and financial burden associated with the harms inherent in the use of inappropriate medications in elderly population is better avoided by provision of a clinical service such as easily accessible. In addition, 3.6% of the current population translates to about120,000 elderly patients in Qatar (Estimated pop is about 3million). Hence, there is a lot of potential inherent benefits associated with the deployment of a care-enhancing clinical service such as deprescribing of inappropriate medications in one of the most vulnerable patient group such as the elderly. In addition, it is generally well established that early deployment of interventions focused improving patient outcomes and strengthening the health system often result lasting positive impact on public health.

This sort of explanation benefits from being included in the manuscript. Further, currently it would appear to be illegal for community pharmacists to do unless and until legislative change is effected, so how quickly can community deprescribing be instituted legally and effectively?. Further, in the 3.6% of the population, how many may be managed by hospital pharmacists who are trialling deprescribing already?

Lines 63-65 assert that the greatest risk to the elderly from medication lies in the number of medicines, and certainly, the more medicines taken, the more there can be a risk of drug-drug, drug-herb, drug-disease state interactions, but there are also some medicines that have an inherent risk of adverse effects in the elderly, even with a lesser number of medications taken. These include NSAIDS, medicines with an anticholinergic effect, some diuretics, antihypertensives and others (https://www.msdmanuals.com/en-au/professional/geriatrics/drug-therapy-in-older-adults/drug-categories-of-concern-in-older-adults).

So some additional clarity in the definition about how medicines may be harmful as medicines have many different perceptions to lay people, to various health professionals such as doctors, specialist doctors, nurses, pharmacists.

Lines 102-104. The manuscript states, “Indeed, only one such study focused on the assessment of community pharmacists’ knowledge of deprescribing and their self-perceived enablers and barriers to providing the service in the United Arab Emirates was identified.”

It can be helpful to identify the search criteria utilised as the databases, the search terms (e.g. whether words were used or truncation was included) and the timeframe as search engines often give different outcomes at different times, so when the search occurred also matters.

METHODS: The requested information in PLOS ONE is detailed and clear – the journal requires sufficient detail to allow suitably skilled researchers to fully replicate this study. When the methods are well established, authors may cite articles where those are described in detail, but even so THIS manuscript should include sufficient information to be understood independent of those references.

Please provide a copy of the survey in an appendix so format, item wording, response sets, item order can be considered.

ANALYSIS: The manuscript identifies that IBM SPSS (Statistics for Windows, Version 29.0. Armonk, NY: IBM Corp.) software was used for data analysis. Descriptive statistics such as median (IQR) was used for continuous data with non-normal distribution while frequency (%) was used for categorical data, with significance set at p ≤ 0.05.

Additional clarity sought: In a manuscript, it is beneficial to identify those functions within SPSS that were used to analyse the data. In survey research, it usually matters both what you ask and how you ask it. So, for example, the manuscript data was stated to have been analysed to yield descriptive statistics such as median (IQR) for continuous, non-normal data and frequency for categorical data.

The SPSS Descriptive function data analysis can yield mean, sum, standard deviation, variance, minimum, maximum, range, skewness, kurtosis and standard error of the mean.

The SPSS Frequencies function yields mean, standard deviation, variance, minimum, maximum, range, skewness, kurtosis, valid & missing responses, median, mode quartiles, percentiles.

It is beneficial to identify the SPSS function actually used as that assists a reader to see whether that data analysis could yield the results of the study or whether a different analysis may have yielded more relevant or more useful result.

Words discussing the analysis can also assist readers as can the actual wording of items, order of items, response set and format of the survey, the study instrument.

The manuscript identified that it elicited responses on a Likert-like response set. Most commonly Likert scales have two extreme responses, two moderate responses and a neutral response per item (strongly disagree, disagree, neutral, agree, strongly agree or reverse order).

Subsequently, the responses were described as ranked, “Further, study participants were asked to rank their responses to the items in Section B and C on a 5-point Likert-type scale (highest =5, high = 4, moderate= 3, low = 2, Lowest = 1)”. The difference between rating and ranking lies in the information elicited. Rating scale questions such as Likert scales seek to elicit respondents’ attitude(s) towards something ( how much do you agree with this statement, for example). A ranking question seeks a respondent’s preference order.

More detailed information is needed to properly understand the results obtained and the conclusions drawn, so please provide additional information, and in particular, more detail.

RESULTS: PLOS ONE accepts single categories across Results/Discussion/Conclusions or merged categories such as Results/Discussion or Discussion/ Conclusions. This manuscript presents all three sections separately. Only Results will be addressed for review in this section.

The Results section starts with a presentation of the characteristics of the respondents noting for example, that “… [they] consisted mainly of 5 nationalities (91.0%; 182/200) including Indian (40.5%), Egyptian (22.5%), Jordanian (22%) and Sudanese (6%). Since only 182/200 may have responded to this item the absence of Qatari respondents was of interest as in 2024 it might be expected that there could be perhaps some 12 cohorts of graduating students from Qatar University since the course is listed as commencing in 2007 with full accreditation in 2011 (https://www.qu.edu.qa/static_file/qu/colleges/pharmacy/documents/Pharmacy%20A4%20Magazine-compressed%2019AY.pdf).

Additional detail and context would be helpful to fully understand the results of the study and its outcomes. It would be beneficial to understand where Qatari educated pharmacists work – data were elicited from chain pharmacies and none were identified as Qatari, so perhaps Independent pharmacies? Industrial pharmacy? Hospital pharmacy? Further, in a 2021 study which included independent and chain pharmacies differences were noted such as a preponderance of males rather than the females majority identified in the current study.

Response-22: We respectfully submit that the results obtained are as presented in the manuscript and the observed trend is consistent with published reports from several studies from Qatar. The community pharmacy sector in Qatar is dominated mainly by foreigners.

Firstly, then please provide the context using the published reports from several studies in Qatar cited in your reply as response 22. Irrespective, pharmacists working in Qatar would seem to be bound by the Law irrespective of nation of initial education and registration/authorisation to practice. Further, it is of interest to pharmacists/researchers globally where Qatari educated pharmacists work and WHY community pharmacy is run by foreigners, which may often be considered an issue in some nations. PLOS is an international journal and while pharmacy has a role in most if not all nations, in my experience, the roles, ownership, ability to practice as an out-of-nation educated pharmacist vary, sometimes greatly.

DISCUSSION: The discussion identified that the community pharmacist respondents would find continuing professional development, the availability of a shared electronic patient record and an improved remuneration package to be key enabling factors in deprescribing in older adults in Qatar who currently reported to be 3.6% of the population. However, there are already reports of an initiative in outpatient pharmacies at Rumailah Hospital, Qatar (Alyazeedi, A., Sherbash, M., Algendy, A. F., Stewart, C., Soiza, R. L., Alhail, M., Aldarwish, A., Stewart, D., Awaisu, A., Ryan, C., & Myint, P. K. (2024). Enhancing Medication Safety through Implementing the Qatar Tool for Reducing Inappropriate Medication (QTRIM) in Ambulatory Older Adults. Healthcare (Basel, Switzerland), 12(12), 1186. https://doi.org/10.3390/healthcare12121186). If hospital outpatient pharmacies in 2022-23 are using, and others perhaps intending to use Qatar Tool for Reducing Inappropriate Medication (QTRIM) in Ambulatory Older Adults i.e. community dwelling, what is the rationale for community pharmacists to also undertake deprescribing? This considers the significance of the outcomes of the manuscript study compared to an existing tool and implementation in outpatient pharmacies.

Please discuss and contextualise.For example, how many elderly people see only community pharmacists, how many or what proportion see both a hospital pharmacist and a community pharmacist and how many or what proportion see only a hospital pharmacist? Some of the reasons that detail and information is important to understanding is what others say about pharmacy practice in Qatar. For example, Nadir Kheir in a book chapter in December 2016 entitled "Pharmacy Practice in Qatar" identified that pharmacy services in Qatar are rapidly developing and pharmacists are rapidly changing, committing to lifelong learning, self leadership and self development. Further, within that chapter it was observed that one of the strongest drivers of the change and development in pharmacy practice is Qatar's National Health Strategy which a further on community pharmacy. It was envisioned that community pharmacy could contribute to a primary health care model of practice and care. So, from 2016 to 2024, what has happened?

LIMITATIONS: Limitations section identifies those characteristics of the study methodology or design that impact the research outcomes or interpretations. As such the limitations are addressed in the section from lines 294 to 301. The limitations identified were firstly, non-probability sampling as participants were drawn predominantly from chain or banner group pharmacies which represent 75% approximately of the community pharmacies in Qatar.

So, this is a valid issue and benefits from discussion as to any ameliorating factors or processes that the research team may have applied. It was also not known how community pharmacies are owned as the owners’ views and actions may also influence employed pharmacists’ perceptions. Context is important as there is not one single vision and practice of pharmacy, or even community pharmacy, world-wide. How different might community pharmacy be in New Zealand, in Thailand, in Japan, in Spain, in Sweden, in Brazil, for example? In some countries, pharmacies may only be owned by registered pharmacists, sometimes with limits imposed on the number of pharmacies owned by a registered pharmacist, in some countries companies can own pharmacies and employ pharmacists to provide services, so context benefits from greater clarity about the situation in Qatar.

The second limitation noted was that a quantitative survey design may have triggered social desirability bias or social acquiescence though the link between methodology (survey) and favourable responses is not clearly explained but benefits from being so.

To my perception, this is a well published area - there are a number of approaches such as published articles that consider this and also social desirability scales which could be included in the study instrument, and other options that can be explored such as Lechner and colleagues 2019; Bergen and Labonte 2020; Larson 2019; Primi and colleagues 2019; Kreitchmann and colleagues 2019; Tan and colleagues 2022 . Which approaches did the researchers use? If none of the approaches were used what was the rationale for not doing so?

So please, provide the critical analyses, the detail, the context that enhance understanding of your study, its results and proposed outcomes.

Reviewer #2: The manuscript looks good and I agree with proceeding for publication, but small comment; the sentence in the line 99-101, I did not any references of the studies that were mentioned

7. PLOS authors have the option to publish the peer review history of their article (what does this mean?). If published, this will include your full peer review and any attached files.

Reviewer #1: No

Reviewer #2: **Yes: **MOHAMMED ABDULLAH KUBAS

---

## [Author Response · Author response to Decision Letter 1]

7 Dec 2024

07 December 2024

The Editor-In-Chief

PLOS ONE

Dear Sir,

Re: Manuscript #PONE-D-24-26224R1 – “Enablers and barriers to community pharmacists’ readiness to implement deprescribing of inappropriate medications for older adults in a developing setting”

We are sincerely grateful for the opportunity to revise, for the second time, the manuscript #PONE-D-24-26224R1, titled “Enablers and barriers to community pharmacists’ readiness to implement deprescribing of inappropriate medications for older adults in a developing setting” which is under your consideration for publication in the PLOS ONE. We thankful for the additional comments and useful suggestions provided by the reviewers. Please find stated below our point-by-point response to the reviewers’ comments. 

REVIEWER’S COMMENTS

Reviewer #1: 

• Comment-1: Currently those in Qatar older than sixty years of age constitute 3.6% of the population though that is predicted to increase to approximately 20.3% by 2050, which is in 26 years’ time, so lacking apparent immediate urgency. So, please seek to firmly and clearly identify the significance or implications of undertaking this study now in 2024 - Why is it needed? Why now? What has to change and what else has to happen for the activity on which this research is focused to come into effect. In my experience, often authors may be well aware of what is happening in their setting, what could be desirable and will of the pharmacist population to provide a proposed service or modify a standard procedure, for example. However, readers from outside the setting are often quite unfamiliar with a study's setting, and in lieu of more information may wonder if it is much the same as in the reader's setting. In my experience providing context, information, including some statistics will assist the reader the view the study within the actual study setting.

• Response-1: We are thankful once again for this important observation and we see the points raised by the reviewer. However, we thought we have addressed this concerns in our previous response. It is generally well-documented that prevention is not only better but it is more cost-effective than cure. Hence, we are of the opinion that the potential clinical and financial burden associated with the harms inherent in the use of inappropriate medications in elderly population is better avoided by provision of deprescribing service especially at the primary care level where community pharmacists are easily accessible. In addition, 3.6% of the current population translates to about 120,000 elderly patients in Qatar (Estimated pop is about 3million). Hence, there is a lot of potential inherent benefits associated with the deployment of deprescribing of inappropriate medications in one of the most vulnerable patient group such as the elderly without having to wait unnecessarily until 2050. In addition, it is generally well established that early deployment of interventions focused improving patient outcomes and strengthening the health system often result lasting positive impact on public health. 

• Comment-2: Issue - A little more clarity would beneficial - what is a dispassionate review? What benefit arises from a dispassionate view? Why not a standard protocol? Why not take into consideration the patient’s perceptions of their medications or issues they may be having? Further, the key is to identify medications that are no longer required - for which reasons? duplication of therapy? continuation of medication to manage an acute condition? removal of a medication that manages an adverse effect of another medication taken by the patient? Issue- benefit of additional clarity as to why deprescribing needs to be added, legally, to community pharmacists’ scope, and why the timing since hospital pharmacists have and are trialing a deprescribing initiative- how is it that deprescribing is within scope for hospital pharmacists in Qatar but not community pharmacists?

Response-2: Heartfelt thanks to the reviewer for this additional comment. We submit with all due respect that the word “dispassionate” is self-explanatory. However, we have replaced this word with a more appropriate one as suggested [Pg 4, line 64]. In addition, we thank the reviewer for all the other issues that was raised in the review comment. However, we submit with all due respect these issues are outside the scope of the objectives of the current study. However, we believe that they are good leads for further research, and we are thankful to the reviewer. 

Comment-3: Lines 63-65 assert that the greatest risk to the elderly from medication lies in the number of medicines, and certainly, the more medicines taken, the more there can be a risk of drug-drug, drug-herb, drug-disease state interactions, but there are also some medicines that have an inherent risk of adverse effects in the elderly, even with a lesser number of medications taken. These include NSAIDS, medicines with an anticholinergic effect, some diuretics, antihypertensives and others (https://www.msdmanuals.com/en-au/professional/geriatrics/drug-therapy-in-older-adults/drug-categories-of-concern-in-older-adults). So some additional clarity in the definition about how medicines may be harmful as medicines have many different perceptions to lay people, to various health professionals such as doctors, specialist doctors, nurses, pharmacists.

Response-3: Heartfelt thanks to the reviewer for this excellent suggestion. We concur and the manuscript has been revised accordingly [Pg 3, line 61-63].

• Comment-4: Lines 102-104. The manuscript states, “Indeed, only one such study focused on the assessment of community pharmacists’ knowledge of deprescribing and their self-perceived enablers and barriers to providing the service in the United Arab Emirates was identified.”

It can be helpful to identify the search criteria utilised as the databases, the search terms (e.g. whether words were used or truncation was included) and the timeframe as search engines often give different outcomes at different times, so when the search occurred also matters. 

Response-4: Many thanks to the reviewer for the suggestion. However, the study was not a systematic review and hence we are of the opinion that the information presented about the procedure used for the literature review suffices.

• Comment-5: METHODS: The requested information in PLOS ONE is detailed and clear – the journal requires sufficient detail to allow suitably skilled researchers to fully replicate this study. When the methods are well established, authors may cite articles where those are described in detail, but even so THIS manuscript should include sufficient information to be understood independent of those references. Please provide a copy of the survey in an appendix so format, item wording, response sets, item order can be considered., so clarity in this study’s use is beneficial

• Response-5: Heartfelt thanks to the reviewer for this comment. Adequate information has been provided in the methods sections regarding all the procedures used at the various stage of the study, including the development and validation of the data collection tool. The data collection tool is available upon reasonable request.

• Comment-6: Lines 63-65 assert that the greatest risk to the elderly from medication lies in the number of medicines, and certainly, the more medicines taken, the more there can be a risk of drug-drug, drug-herb, drug-disease state interactions, but there are also some medicines that have an inherent risk of adverse effects in the elderly, even with a lesser number of medications taken. These include NSAIDS, medicines with an anticholinergic effect, some diuretics, antihypertensives and others (https://www.msdmanuals.com/enau/professional/geriatrics/drug-therapy-in-older-adults/drug-categories-of-concern-in-older-adults).

• Response-6: Sincere thanks to the reviewer for this excellent comment. We concur with the reviewer point and we think this assertion has already been captured in the manuscript [Line 61-65]. We were conscious of balancing comprehensiveness with conciseness to minimize redundancies.

• Comment-7 ANALYSIS: The manuscript identifies that IBM SPSS (Statistics for Windows, Version 29.0. Armonk, NY: IBM Corp.) software was used for data analysis. Descriptive statistics such as median (IQR) was used for continuous data with non-normal distribution while frequency (%) was used for categorical data, with significance set at p ≤0.05. Additional clarity sought: In a manuscript, it is beneficial to identify those functions within SPSS that were used to analyse the data. In survey research, it usually matters both what you ask and how you ask it. So, for example, the manuscript data was stated to have been analysed to yield descriptive statistics such as median (IQR) for continuous, non-normal data and frequency for categorical data. The SPSS Descriptive function data analysis can yield mean, sum, standard deviation, variance, minimum, maximum, range, skewness, kurtosis and standard error of the mean. The SPSS Frequencies function yields mean, standard deviation, variance, minimum, maximum, range, skewness, kurtosis, valid & missing responses, median, mode quartiles, percentiles. It is beneficial to identify the SPSS function actually used as that assists a reader to see whether that data analysis could yield the results of the study or whether a different analysis may have yielded more relevant or more useful result. Words discussing the analysis can also assist readers as can the actual wording of items, order of items, response set and format of the survey, the study instrument. The manuscript identified that it elicited responses on a Likert-like response set. Most commonly Likert scales have two extreme responses, two moderate responses and a neutral response per item (strongly disagree, disagree, neutral, agree, strongly agree or reverse order). Subsequently, the responses were described as ranked, “Further, study participants were asked to rank their responses to the items in Section B and C on a 5-point Likert-type scale (highest =5, high = 4, moderate= 3, low = 2, Lowest = 1)”. The difference between rating and ranking lies in the information elicited. Rating scale questions such as Likert scales seek to elicit respondents’ attitude(s) towards something (how much do you agree with this statement, for example). A ranking question seeks a respondent’s preference order. More detailed information is needed to properly understand the results obtained and the conclusions drawn, so please provide additional information, and in particular, more detail.

Response-7: We thank the reviewer for these comments. We respectfully submit that adequate information has been provided with sufficient clarity in the methods sections regarding the data analyses conducted in the study [Line 141-184]. We are of the opinion that this suffices and it is consistent with the established norm in the scholarly world. 

• Comment-8: RESULTS: PLOS ONE accepts single categories across Results/Discussion/Conclusions or merged categories such as Results/Discussion or Discussion/ Conclusions. This manuscript presents all three sections separately. Only Results will be addressed for review in this section. The Results section starts with a presentation of the characteristics of the respondents noting for example, that “… [they] consisted mainly of 5 nationalities (91.0%; 182/200) including Indian (40.5%), Egyptian (22.5%), Jordanian (22%) and Sudanese (6%). Since only 182/200 may have responded to this item the absence of Qatari respondents was of interest as in 2024 it might be expected that there could be perhaps some 12 cohorts of graduating students from Qatar University since the course is listed as commencing in 2007 with full accreditation in 2011 (https://www.qu.edu.qa/static_file/qu/colleges/pharmacy/documents/Pharmacy%20A4%20Magazine-compressed%2019AY.pdf). Additional detail and context would be helpful to fully understand the results of the study and its outcomes. It would be beneficial to understand where Qatari educated pharmacists work – data were elicited from chain pharmacies and none were identified as Qatari, so perhaps Independent pharmacies? Industrial pharmacy? Hospital pharmacy? Further, in a 2021 study which included independent and chain pharmacies differences were noted such as a preponderance of males rather than the females majority identified in the current study. Response-22: We respectfully submit that the results obtained are as presented in the manuscript and the observed trend is consistent with published reports from several studies from Qatar. The community pharmacy sector in Qatar is dominated mainly by foreigners. Firstly, then please provide the context using the published reports from several studies in Qatar cited in your reply as response 22. Irrespective, pharmacists working in Qatar would seem to be bound by the Law irrespective of nation of initial education and registration/authorisation to practice. Further, it is of interest to pharmacists/researchers globally where Qatari educated pharmacists work and WHY community pharmacy is run by foreigners, which may often be considered an issue in some nations. PLOS is an international journal and while pharmacy has a role in most if not all nations, in my experience, the roles, ownership, ability to practice as an out-of-nation educated pharmacist vary, sometimes greatly.

• Response-8: We are thankful to the reviewer for this comment and we see the point. However, we thought we have addressed this in our previous response. In addition, we respectfully submit that some of the suggested corrections are outside the scope of the stated study objectives. We sincerely appreciated the reviewer’s generosity in identifying these potential leads for further research but we desirous of ensuring that the manuscript is laser-focused on the stated objectives. 

Comment-9 - DISCUSSION: The discussion identified that the community pharmacist respondents would find continuing professional development, the availability of a shared electronic patient record and an improved remuneration package to be key enabling factors in deprescribing in older adults in Qatar who currently reported to be 3.6% of the population. However, there are already reports of an initiative in outpatient pharmacies at Rumailah Hospital, Qatar (Alyazeedi, A., Sherbash, M., Algendy, A. F., Stewart, C., Soiza, R. L., Alhail, M., Aldarwish, A., Stewart, D., Awaisu, A., Ryan, C., & Myint, P. K. (2024). Enhancing Medication Safety through Implementing the Qatar Tool for Reducing Inappropriate Medication (QTRIM) in Ambulatory Older Adults. Healthcare (Basel, Switzerland), 12(12), 1186. https://doi.org/10.3390/healthcare12121186). If hospital outpatient pharmacies in 2022-23 are using, and others perhaps intending to use Qatar Tool for Reducing Inappropriate Medication (QTRIM) in Ambulatory Older Adults i.e. community dwelling, what is the rationale for community pharmacists to also undertake deprescribing? This considers the significance of the outcomes of the manuscript study compared to an existing tool and implementation in outpatient pharmacies. Please discuss and contextualize. 

Response-9 - Many thanks to the reviewer for this comment. This comment has been addressed in our previous response as stated below, and we believe that this suffices:

“The study alluded to was focused on ambulatory patients in a hospital setting, while the current study is focused on community pharmacists in the community setting. The rationale for community pharmacist to undertake deprescribing is well-documented and has been alluded to in the manuscript [Line 68-79].

Comment-10: For example, how many elderly people see only community pharmacists, how many or what proportion see both a hospital pharmacist and a community pharmacist and how many or what proportion see only a hospital pharmacist? Some of the reasons that detail and information is important to understanding is what others say about pharmacy practice in Qatar. For example, Nadir Kheir in a book chapter in December 2016 entitled "Pharmacy Practice in Qatar" identified that pharmacy services in Qatar are rapidly developing and pharmacists are rapidly changing, committing to l

---

## [Editor Report · Decision Letter 2]

10 Dec 2024

Enablers and barriers to community pharmacists’ readiness to implement deprescribing of inappropriate medications for older adults in Qatar

PONE-D-24-26224R2

Dear Dr. Yusuff,

We’re pleased to inform you that your manuscript has been judged scientifically suitable for publication and will be formally accepted for publication once it meets all outstanding technical requirements.

Kind regards,

Jenny Wilkinson, PhD

Academic Editor

PLOS ONE
---

## [Editor Report · Acceptance letter]

9 Jan 2025

PONE-D-24-26224R2 

PLOS ONE

Dear Dr. Yusuff, 

I'm pleased to inform you that your manuscript has been deemed suitable for publication in PLOS ONE. Congratulations! Your manuscript is now being handed over to our production team.

Kind regards, 

on behalf of

Dr Jenny Wilkinson 

Academic Editor

PLOS ONE